# Assessing Neurogenic Lower Urinary Tract Dysfunction after Spinal Cord Injury: Animal Models in Preclinical Neuro-Urology Research

**DOI:** 10.3390/biomedicines11061539

**Published:** 2023-05-26

**Authors:** Adam W. Doelman, Femke Streijger, Steve J. A. Majerus, Margot S. Damaser, Brian K. Kwon

**Affiliations:** 1International Collaboration on Repair Discoveries, University of British Columbia, Vancouver, BC V5Z 1M9, Canada; doelman@icord.org (A.W.D.); streijger@icord.org (F.S.); 2Department of Electrical, Computer and Systems Engineering, Case Western Reserve University, Cleveland, OH 44106, USA; sjm18@case.edu; 3Advanced Platform Technology Center, Louis Stokes Cleveland VA Medical Center, Cleveland, OH 44106, USA; damasem@ccf.org; 4Department of Biomedical Engineering, Lerner Research Institute, Cleveland Clinic, Cleveland, OH 44195, USA; 5Department of Orthopaedics, Vancouver Spine Surgery Institute, University of British Columbia, Vancouver, BC V5Z 1M9, Canada

**Keywords:** spinal cord injury, animal models, cystometry, urodynamics, neuro-urology, neurogenic lower urinary tract dysfunction

## Abstract

Neurogenic bladder dysfunction is a condition that affects both bladder storage and voiding function and remains one of the leading causes of morbidity after spinal cord injury (SCI). The vast majority of individuals with severe SCI develop neurogenic lower urinary tract dysfunction (NLUTD), with symptoms ranging from neurogenic detrusor overactivity, detrusor sphincter dyssynergia, or sphincter underactivity depending on the location and extent of the spinal lesion. Animal models are critical to our fundamental understanding of lower urinary tract function and its dysfunction after SCI, in addition to providing a platform for the assessment of potential therapies. Given the need to develop and evaluate novel assessment tools, as well as therapeutic approaches in animal models of SCI prior to human translation, urodynamics assessment techniques have been implemented to measure NLUTD function in a variety of animals, including rats, mice, cats, dogs and pigs. In this narrative review, we summarize the literature on the use of animal models for cystometry testing in the assessment of SCI-related NLUTD. We also discuss the advantages and disadvantages of various animal models, and opportunities for future research.

## 1. Background

Neurogenic lower urinary tract dysfunction (NLUTD) is a common secondary complication of traumatic spinal cord injury (SCI), and the recovery of bladder function has been reported as a key priority, particularly in those with paraplegia [1,2,3,4,5,6]. Approximately 80% of SCI individuals develop some degree of lower urinary tract (LUT) dysfunction within the first year following injury, which has a significant impact on their quality of life [7]. NLUTD as a result of traumatic SCI can include a multitude of conditions, some of which can be life-threatening. Accurate and reliable assessment techniques are therefore necessary to characterize NLUTD in individuals with SCI and guide their bladder management [8].

Repeated urodynamic studies (UDS) are an essential aid in managing the evolving nature of bladder dysfunction, and urodynamics is presently the ‘gold standard’ for the assessment of NLUTD after SCI. Cystometry, a part of UDS, uses multiple-pressure catheters and retrograde bladder filling for the assessment of the pressure/volume relationship of the bladder during storage and emptying, as well as measures of bladder capacity, sensation and bladder compliance. Cystometry is a provocative test used to elicit features that can be used to identify subjects at risk of urological complications, including renal dysfunction, UTIs and incontinence [9,10,11,12]. A key feature of NLUTD is neurogenic detrusor overactivity (NDO) [13], in which the detrusor muscle contracts spontaneously during bladder filling. Another aspect of NLUTD is detrusor sphincter dyssynergia (DSD), in which the external urethral sphincter (EUS) contracts, rather than relaxes, during voiding [14]. NDO with or without DSD is a common urological complication following SCI [15]. Impaired compliance, reduced bladder capacity, or high detrusor leak point pressure (DLPP), which is defined as the lowest vesical pressure at which urine leaks from the bladder during increased abdominal pressure, such as in straining or coughing, in the absence of a detrusor contraction, can identify a patient at high-risk of upper urinary tract deterioration (UUTD) [16]. The long-term persistence of NLUT symptoms can lead to impaired bladder capacity, incontinence, inefficient voiding with high post-void residual volumes, and a high risk of UUTD that can be life threatening [17,18,19]. Therefore, the prompt detection of high-risk features and timely intervention, treatment and management are crucial [8].

The fundamental exploration of these pathologies and the investigation of the precise underlying mechanism(s) are often not possible in human subjects. Moreover, the quantitative observation of urinary tract function before and after SCI—in the same individual—is effectively impossible with human patients. Animal models are therefore critical for investigating the in vivo consequences of NLUTD, and they allow for the development and testing of novel therapeutic approaches and understanding the complex pathophysiology of a neurogenic bladder [20]. Given the importance of animal models, clinical UDS has been adapted to a wide variety of animal species. Historically, mice, rats, and cats have been the most commonly used species for UDS investigations after SCI [21]. Their small size, however, does not permit the evaluation of urological devices also designed for human use. More recently, the minipig has been proposed as a viable intermediary, large animal model for translational studies in NLUTD due to its physiological similarities to humans [22]. The present review summarizes the literature on various animal models used to characterize NLUTD after SCI. By highlighting the pros and cons of the different animal models and the typical cystometric findings after SCI, we hope that this review will assist the research community engaged in investigating NLUTD in animal models of SCI.

## 2. Literature Review

An electronic search of Pubmed and Google Scholar for animal models of SCI was performed, using the following terms: lower urinary tract dysfunction OR neurogenic lower urinary tract dysfunction AND spinal cord injury AND animal models. The titles and abstracts of studies of the resultant articles were screened. Reviews, case reports, conference proceedings, letters to the editor, and stand-alone abstracts were excluded. The full texts of the remaining manuscripts were retrieved. Original studies that had used animal models of SCI incorporating functional bladder assessment after SCI were included. Neurological injury and impairment not pertaining to SCI, such as, but not limited to, ventral root avulsion, peripheral nerve crush, lumbar canal stenosis, traumatic brain injury, multiple sclerosis, cauda equina syndrome, and occult neural tube defect, were excluded. In vitro and ex vivo preparations and studies that contained no functional assessment of LUT function were excluded. A manual cross-reference search of the selected articles was conducted to avoid missing relevant publications.

## 3. Animal Cystometry

In many ways, animal cystometry mimics clinical testing and involves placing a catheter into the bladder and recording cystometric parameters during the artificial retrograde filling of the bladder. However, unlike human cystometric assessment, which involves transurethral catheterization into an unsedated subject and observing one to two fill-empty cycles, several experimental parameters can be modified in animal studies to best suit the proposed model and address the research question(s). Alterations to conventional cystometry can include different catheterization techniques, anesthetic or conscious preparations, and filling strategies.

In animals, catheters can be placed acutely or be indwelling in order to permit chronic urodynamic assessment. They can be inserted either transurethrally or via a suprapubic approach, with the latter requiring surgical preparation in an anesthetized animal [23]. Acute surgical stress and pain can activate CNS pathways to release stress hormones and may influence LUT function [24,25], which must be considered when performing cystometry immediately following acute invasive catheter placement. Telemetric pressure sensors have also been utilized to permit chronic assessment without the externalization of tubing [26,27].

Animal cystometry can be carried out in conscious, anesthetized or decerebrate states. Given the considerable difficulty involved in conducting cystometry in awake animals, anesthetized preparations are common. Anesthetic agents regularly used in animal cystometry before and after SCI include urethane [28], isoflurane [29,30], dexmedetomidine [31], ketamine [32,33], and others. Anesthetic agents, particularly inhalants, have been shown to affect LUT function by repressing the micturition reflex [34]. In contrast, conscious preparations avoid the use of anesthetics but often require physical restraint, which has also been shown to alter voiding function and reduce voided volumes [35,36]. Conscious activity in unrestrained ambulatory conditions is therefore more physiological, however, and considerable noise and motion artifact can complicate the interpretation of the resultant data [28]. Decerebrate preparations permit the assessment of reflex micturition without the effect of anesthetics or motion artifact, and have been shown to preserve voiding efficiency with low residual volumes in rodents [37,38,39]. However, in rats, decerebration reduced the bladder capacity by approximately 40% compared to conscious restrained animals [28].

Filling strategies during animal cystometry can include single fill, continuous fill or both. Single filling cystometry is performed by infusing saline into the empty bladder to artificially create a single fill-and-void cycle [31,40,41,42,43,44]. The infusion is stopped at the beginning of a micturition contraction and the fluid voided is collected and measured. The bladder is then emptied to measure residual volume. This approach of eliciting a single fill-and-void cycle is most similar to the manner in which human cystometry is conducted, and thus may be considered more translatable. Alternatively, continuous cystometry is performed using a constant infusion of fluid into the bladder to elicit repetitive voids, which allows the collection of data for a number of voiding cycles [45]. Filling is paused briefly during continuous cystometry at the initiation of voiding for volume collection and residual assessment, and is then resumed to examine multiple voiding cycles in a given session. This is useful to determine whether a drug has effects on the intraluminal pressure (e.g., opening, voiding, or closing pressures during bladder contraction) and volume (i.e., functional bladder capacity, residual urine and voiding efficiency). A faster infusion rate is often chosen for continuous filling cystometry than for single filling cystometry (ex. 0.21 mL/min vs 0.04 mL/min in rats), especially when multiple doses of drugs are being tested in cumulative dose-response studies [41,42]. Combined single-fill and continuous cystometry may also be used by performing continuous filling after a single-fill cycle in the same preparation [46].

## 4. Cystometric Findings in Animal Models of SCI

NLUTD has been investigated in a variety of animal models of SCI, with the majority of studies inducing the SCI in the thoracic spine by complete transection, but with other injury mechanisms such as contusion, compression and heat ablation also described (Figure 1). While cats and dogs were commonly utilized in the first investigations into the basic neural mechanisms underlying LUT function in the late 19th century [47] and late 1960–1970s [48,49,50,51], more recent studies focusing on NLUTD after SCI have been performed using rats [36,40,44,52,52,53,54,55,56,57,58,59,60,61,62,63,64,65,66,67,68,69,70,71,72,73,74,75,76,77,78,79,80,81,82,83,84,85,86,87,88,89,90,91,92,93,94,95,96,97,98,99,100,101,102,103,104,105,106,107,108,109,110,111,112,113,114,115], mice [35,116,117,118,119,120,121,122], cats [29,30,32,33,48,123,124,125,126,127,128,129,130,131,132,133,134,135,136], dogs [44,51,137,138,139,140], and pigs [31,141] (Table 1). Although the use of each species presents advantages and disadvantages, there are several common cystometric features that will be discussed in the following sections.

### 4.1. Mice and Rats

Small rodents, such as rats and mice, have formed the foundation of our understanding of the neural control of micturition and the mechanisms underlying the impairment of LUT function after SCI, in part because their bladders have been well-characterized by both in vitro and in vivo experiments [20,123,142,143,144,145]. Their lifespan of about 1–2 years means that it is also feasible to evaluate the effects of chronic SCI [118]. The availability of several transgenic mouse models can provide valuable mechanistic insights into LUT function [146,147,148].

To conduct rodent cystometry, catheterization can be performed transurethrally [69,149], as well as by using either acute [55,64,150,151,152] or chronic [153] suprapubic approaches. Suprapubic insertion is common in these models given the small urethral size for insertion, and because it reduces urethral and EUS irritation during EMG recording [154]. While bladder activity has been observed 2 days after catheter implantation in rodents, it often does not return to pre-implantation levels until ~6 days after implant [155,156,157,158], demonstrating that the timing of experiments following chronic catheter implantation must be considered. Needle electrodes are most often utilized to assess EUS EMG activity in mice and rats, due to their smaller size compared to the patch electrodes used for large animals such as cats, dogs, and pigs [154]. In rodents, the striated muscle of the proximal urethra plays an important role in the micturition process [159,160]. Unlike human EUS function, rodent micturition is mediated by the contraction of the detrusor accompanied by coordinated, intermittent contractions of the striated muscle of the EUS, with frequency bursts of 4–6 Hz [60,69,123,145,161]. The EUS EMG silent/active period duration can thus be used as an outcome metric for the assessment of pathological EUS function, relative to baseline and compared following therapeutic intervention.biomedicines-11-01539-t001_Table 1Table 1Summary of filling cystometry studies in animal models of SCI. Among the 112 studies included, 98 characterized LUT dysfunction related to experimental SCI through filling cytometry. (SCI = spinal cord injury; Ref = reference #; No cath = no catheter; T = thoracic; L = lumbar; S = sacral; EMLA = lidocaine 2.5% and prilocaine 2.5%; d = days; w = weeks; α-chlor. = alpha-chloralose).SpeciesMechanismLevelCatheterTime Post-SCIAnesthetic UsedRef.MouseTransectionT8Transurethral18 weeksIsoflurane[35]T8-9Suprapubic2 weeks
[121]T8-9Suprapubic2 weeks
[119]T8-9Suprapubic2–4 weeksEMLA[118]T8-9Suprapubic4 weeksELMA[116]T8-9Suprapubic4 weeksIsoflurane[162]T8-9Suprapubic4 weeks
[120]T8-9Suprapubic4 weeks
[122]T8-9Suprapubic4 weeks
[117]Rat & MouseTransectionT8-9Suprapubic4 weeks
[36]RatCompressionT8No cath10 weeksIsoflurane[100]T8-T9Suprapubic2, 4 weeks
[75]ContusionL5-S2Suprapubic2–8 weeks
[109]T10corpus spongiosum3–21 days
[86]T10No cath2–3 d–2 weeksIsoflurane[85]T10No cath4 weeks
[56]T10Suprapubic4 weeksUrethane[59]T10Transurethral4 weeksNot specified[115]T8Suprapubic2–3 d–2 weeks
[71]T8Suprapubic2–4 monthsUrethane[114]T8Transurethral8 weeks
[69]T8Transurethral8 weeks
[72]T8-T9Suprapubic2 weeksUrethane[111]T9-T10Suprapubic2 weeks
[108]T9-T10Suprapubic4 d, 2–8 weeksIsoflurane[93]T9-T10Suprapubic5 weeksKetoprofen[112]Heat InjuryT12Suprapubic30 daysUrethane and α-chlor.[62]TransectionL4-L5Transurethral6 weeksPentobarbital[110]T10Suprapubic3 weeks
[67]T10Suprapubic4 weeks
[95]T10Suprapubic4,5 weeks
[101]T10Suprapubic6 weeks
[44]T10Suprapubic6–8 weeks
[74]T10Suprapubic8–12 weeksUrethane[163]T10Transurethral8 weeksKetamine[164]T11Suprapubic3 weeks
[102]T4Suprapubic3 weeksXylazine/Ketamine[97]T6-T7Suprapubic4 weeks
[79]T7-T9Suprapubic2–3 weeks
[40]T7-T9Suprapubic2–3 weeks
[73]T7-T9Suprapubic6 weeksUrethane[66]T7-T9Suprapubic7 weeksUrethane[88]T8Suprapubic1–4 weeks
[53]T8Suprapubic4 weeks
[54]T8Suprapubic4 weeksUrethane[96]T8Suprapubic6 weeks
[105]T8-9,L3-4,L6-S1Suprapubic4 weeksUrethane[90]T8-T10Suprapubic2–3 weeksHalothane[64]T8-T10Suprapubic4–6 monthsUrethane[57]T8-T9Suprapubic1,3,4 weeks
[52]T8-T9Suprapubic1–2 daysUrethane[77]T8-T9Suprapubic1–4 weeks
[165]T8-T9Suprapubic2 weeksHalothane[70]T8-T9Suprapubic3 weeksHalothane[78]T8-T9Suprapubic3 weeks
[83]T8-T9Suprapubic3 weeks
[98]T8-T9Suprapubic4 weeks
[166]T8-T9Suprapubic4 weeksUrethane[113]T8-T9Suprapubic4 weeksUrethane[107]T8-T9Suprapubic4 weeks
[65]T8-T9Suprapubic4–5 weeksUrethane[61]T8-T9Suprapubic4–5 weeksUrethane[60]T8-T9Suprapubic6 weeksUrethane[167]T8-T9Transurethral6–8 weeksUrethane[58]T8-T9Transurethral7 weeksKetamine[84]T9Suprapubic3–28 days
[89]T9-T11Suprapubic6 weeksHalothane[80]T9-T10Suprapubic1–2 weeksChloral hydrate[104]T9-T10Suprapubic1–2 weeks
[91]T9-T10Suprapubic1–8 weeksUrethane[99]T9-T10Suprapubic2, 4 weeks
[168]T9-T10Suprapubic4 weeks
[92]T9-T10Suprapubic4 weeks
[103]T9-T10Suprapubic4 weeksUrethane[81]T9-T10Suprapubic6 weeksUrethane[68]T9-T10Transurethral1 dayUrethane[87]T9-T10Transurethral1 d, 4 weeksUrethane[76]T9-T10Transurethral4 weeksUrethane[82]T9-T11Suprapubic4–7 weeksUrethane[55]T9-T11Suprapubic6–8 weeksUrethane[94]Transection/CompressionT8Suprapubic6, 14 weeks
[106]Transection/ContusionT8Suprapubic2, 6 weeks
[150]T8Transurethral2 d–2 weeksChloral hydrate[63]RabbitCauderizationT12-L2Transurethral10–12 weeksNot specified[169]TransectionT10Transurethral3 weeks
[170]CatCompressionT1Suprapubic1–2 weeks
[48]T1Suprapubic8–10 weeks
[123]TransectionC6-T1Transurethral3–10 weeks
[124]L1Suprapubic6 h–38 daysHalothane[125]TransectionT10Suprapubic3–6 weeks
[126]T10SuprapubicimmediatelyHalothane[33]T10-T11Transurethral2–6 months
[127]T10-T11Transurethral3–10 months
[128]T11-T12Suprapubic8 weeksIsoflurane + α-chlor.[129]T11-T12Suprapubicnot specified
[130]T12SuprapubichoursKetamine + α-chlor.[131]T12-T13Suprapubic8 weeksIsofluorane + α-chlor.[29]T13Transurethral2–8 weeks
[132]T8-T12Suprapubic/Transurethral2–12 h, 4–14 wUrethane or α-chlor.[32]T9-T10Suprapubic6–8 weeksα-chloralose[133]T9-T10Transurethral18–24 weeks
[30]T9-T10Transurethral4–5 weeksα-chloralose[134]T9-T10Transurethral4–50 weeksα-chloralose[135]Transection/ContusionT8Transurethral3 weeksKetamine/Xylazine.[136]DogTransectionT10Suprapubic/Transurethral1–4 weeksHalothane[51]T10Transurethral1–8 monthsNot specified[137]T8-T9no cathnot specifiedPentobarbital[138]T8-T9Transurethral1–8 weeksnot specified[139]T8-T9Urethral + Ureteral cathnot specifiedPentobarbital[140]T11-T12Transurethral1–6 weeksKetamine[44]PigCompressionT11-T12Transurethral1–16 weeksPropofol/Xylazine[141]ContusionT2/T10Transurethral4–13 weeksDexmedetomidine + atipamezole[31]


Experiments have been performed in anesthetized and conscious conditions. Anesthetic agents such as urethane, ketamine/xylazine, propofol, isofuorane and pentobarbital have all been shown to influence rodent LUT function by varying degrees [34]. Most commonly, higher doses of agents such as propofol and pentobarbital can suppress voiding reflexes, leading to considerable differences in bladder capacity, detrusor contraction pressure, voiding efficiency, and residual volumes between studies [34,171]. Isofluorane is often used as an anesthetic agent during animal cystometry as it is easy to administer, has minimal side effects, and permits a fast recovery; however, it has been reported to suppress bladder contractions with increased inter-contraction intervals in rats [172]. Urethane [58] and ketamine/xylazine [173] are therefore more commonly used in rodent studies as the micturition reflex is maintained in sedated animals. Conscious, restrained experiments such as those developed by Schneider et al. [153] are challenging but have the obvious advantage of avoiding the anesthetic-induced changes to storage and micturition.

After SCI, voiding is severely impaired in rodents, which demonstrate increased bladder capacity, increased maximum detrusor contraction pressure, increased PVR, and decreased voiding efficiency [57,151,174,175]. Tonic dyssynergic EUS activation results in poor voiding efficiency, increased residual volumes, areflexic bladder, and sometimes complete urinary retention [56,73,176]. After SCI, rodent studies have demonstrated an overactive bladder following the emergence of a spinal micturition reflex pathway [143], with decreased voided volume. The loss of EUS activation, NDO, and DSD have also been observed in rodent cystometry studies following supra-sacral spinal cord injury [38,177,178,179,180,181], similar to human SCI. Interestingly, Kadekawa et al. demonstrated differences in voiding after SCI in rats and mice, such that during micturition, SCI rats commonly exhibit EUS-EMG bursting and generate higher voiding efficiencies relative to SCI mice, who generally exhibit no EUS-EMG bursting and poor voiding efficiency after injury [36].

Given that rodents have widespread availability, are relatively cost effective, and have standardized methods for SCI induction and post-SCI care, rodent models are useful as a preclinical model for the study of NLUTD. Rodents have been utilized to test the therapeutic potential of various compounds after SCI, including B-3 adrenoreceptor agonists [117,162], 5′-HT receptor agonists [90,114,163], anti-Nogo-A antibodies [53], α1D-adrenoreceptor agonist [54] and α1DA antagonist [103], s-nitrosoglutathione [108], inosine [106], and immortalized stem cells [75]. Testing in rodents has demonstrated promising results for peripheral nerve transplantation [35], bladder augmentation using acellular matrix grafting [164], detrusor myoplasty [182], hypogastric nerve transection [73], and hypogastric nerve stimulation after pelvic nerve resection [62]. Rodents have been used to test electrical stimulation paradigms at the tibial [52,115], sacral [11,67,98] and pudendal nerves [167]. In addition, they have been used to test the effects of botulinum-A toxin on bladder dysfunction [83,84,96,97]. Rodents have also been used to test bladder management strategies [120], the timing of NDO after SCI [102], the characterization of DSD [68,69,90], as well as to better characterize NLUTD [55,63,64].

In summary, rodents are an excellent model for investigating the fundamental nature of how SCI affects the LUT and the preclinical testing of various therapeutic agents (see Table 2). Their relatively low cost, minimal maintenance requirements and widespread availability have allowed these animals to be extensively studied and well-characterized. Their smaller size does limit the accuracy of some measurements, including urine flow and residual volumes. In addition, the testing of devices intended for clinical application may require extensive miniaturization to be compatible with the rodent model. However, tibial, sacral and pudendal nerve stimulation protocols have been developed for these animals and a vast number of drug, cell and tissue therapies have shown promising results during SCI recovery.

### 4.2. Cats

Historically, cats were among the first species to have their LUT function characterized after SCI and remain a relatively common model for NLUTD, partly due to their large bladder size relative to rodents. Transurethral [133,134,135,136] and suprapubic [30,32,33,131,132] approaches are commonly used to evaluate bladder function in these animals. Under normal conditions, EUS function during storage and voiding in felines more closely resembles human EUS EMG relaxation–contraction such that EUS activity is largely silenced during micturition [183]. In cats, three areas of the brainstem and diencephalon have been shown to be implicated in the control of micturition, including the pontine micturition center, the periaqueductal grey matter and the pre-optic area of the hypothalamus. These regions are also active in humans during micturition, as revealed by fMRI studies [183].

An increased bladder capacity, a decreased amplitude of voiding contractions, a lower voiding efficiency, and a shorter voiding duration occur after SCI in cats [133,184,185]. Reflex bladder contractions induced by bladder distension in SCI cats were shown to be weaker and less efficient than those in intact cats [185]. Detrusor areflexia (i.e., loss of reflex bladder contractions) is common in acute SCI in felines [184]. NDO and DSD have also been reported during filling cystometry in chronic SCI cats [31,55,133,134]. Furthermore, the frequency of non-voiding contractions (sometime referred to as pre-micturition contractions) appears to decrease over time, as demonstrated in chronic transected SCI cats from 4–50 weeks after injury [134], an investigation that has not yet been performed in other models.

Felines have been a useful experimental model for testing therapeutic neuromodulation paradigms after SCI. For instance, Yoo et al. found that selective stimulation of the different branches of the pudendal nerve could evoke excitatory bladder contractions after SCI [186]. The spinal cords of four male cats were surgically transected at T10 and the bladder contractions evoked by the deep pudendal nerve and by cranial sensory nerve stimulation were recorded via a suprapubic catheter. The results revealed the dorsal genital branch of the pudendal nerve that remained intact after spinal cord injury responded to high-frequency stimulation and improved LUT function [186]. Further, Guo et al. found that pudendal neuromodulation using a wearable stimulation device could improve bladder outcomes after SCI in cats by reducing the intravesical pressure and increasing the voiding efficiency 14 weeks after injury [136]. Studies such as these are informative and may facilitate the translation of neuromodulation therapies to elicit or inhibit bladder contractions in SCI individuals. Other methods of sacral neuromodulation have also been assessed in SCI cats [131], along with deep perineal nerve [32], direct bladder [125] and pelvic plexus [132] stimulation. In addition, studies have utilized the feline model to test various therapies to improve the recovery of bladder function after SCI, including 5HT receptor agonists [29,30,133] and autografting the adrenal medulla [130].

The feline model has been commonly used to assess LUT after SCI due to their relatively larger size, availability and established stimulation paradigms. These animals have been valuable for testing various stimulation protocols to elicit and/or inhibit bladder contractions after injury. However, the ethical considerations associated with conducting research in felines can make this model challenging for researchers (see Table 2). Costs associated with housing are also greater than rodents and maintaining this model under anesthesia for extended periods can be considered a limitation in some SCI studies investigating NLUTD.

### 4.3. Dogs

Relatively few studies have utilized dogs as an experimental model to assess LUT function before or after SCI [44,51,138,139,140] and are therefore considerably less well characterized. Under normal conditions, canines exhibit an EUS EMG bursting behavior similar to rodents [139,187].

Complete urinary retention and areflexia of the detrusor for a period of 2–6 weeks occur in dogs acutely after a transection SCI [188,189], similar to acute SCI in people. Levine et al. [190] showed a pattern of increased bladder compliance, capacity, and low maximal bladder pressures from 0–42 days post-injury in dogs with an anatomically incomplete SCI caused by intervertebral disc herniation. During the later stage after SCI, the occurrence of NDO has been detected by cystometry 1 and 3 months after SCI, alongside decreased bladder capacity and compliance [191].

Various approaches to improve bladder function and restore normal continence have been tested in dogs, including transcutaneous bladder stimulation [192], as well as pelvic [193] and pudendal nerve stimulation [191,194] and intraspinal sacral root stimulation [139]. Bladder atony at the time of spinal shock was improved via the functional electrical stimulation of sacral nerve roots [44,140] when delivered to dogs early in the disease process. Hassouna et al. found that fatiguing the EUS via a high-frequency pudendal nerve stimulation blockade, followed by the bladder stimulation of sacral roots, resulted in an increased voiding efficiency in spinal-transected dogs [140]. This method of EUS stimulation fatigue was repeated in a follow-up study by the same group [44], showing reduced EUS EMG activation via the high-frequency stimulation and simultaneous activation of autonomic nerve fibers by low-frequency stimulation. Levine et al. also showed that early pharmacological intervention via the subcutaneous injection of a matrix metalloproteinase inhibitor could improve bladder compliance in dogs with incomplete SCI caused by disc herniation [195].

Dogs have not been extensively studied as it pertains to NLUTD after SCI. Their larger size does permit the testing of novel devices intended for clinical application; however, dogs have been considerably less well characterized and most of the field is moving away from the use of dogs as research animals, likely due to the difficulty in obtaining ethical approval from institutional animal care committees.

### 4.4. Pigs

The anatomical and physiological similarities of the LUT of humans and pigs have been well described in several studies [196,197,198], and pigs have been identified as one of the most clinically relevant models for SCI [22]. Evidence suggests that humans and pigs share many similarities with respect to the LUT, such as size, physiology and, anatomy, and their relatively longer lifespan makes them valuable models for LUT research [199]. Observations of EUS EMG in normal pigs show that, like humans, pigs do not exhibit EUS bursting activity during normal voiding, as seen in rodents [31]. Anatomically, the presence of a horizontal slit-like urethral lumen and the transitional epithelium at the level of the bladder neck are documented anatomical similarities between the pig and human bladder [198]. When comparing bladder size, pigs present a much larger bladder than rodents, as demonstrated in Figure 2, which allows for the testing and development of novel human-sized devices. Although not in the context of SCI, Huppertz and colleagues [27] showed, for example, the feasibility of an implantable pressure system in female Göttingen minipigs as a promising approach by which to facilitate the characterization of bladder function in vivo, and by which to test various therapeutic interventions and techniques in order to inform clinical practice.

Keung et al. characterized NLUTD in Yucatan minipigs after thoracic SCI at the T2 and T10 level [31]. They demonstrated that 68% of SCI minipigs developed NDO 11–17 weeks after contusion/compression injury. The data from a Yucatan minipig before and after contusive SCI during cystometry are shown in Figure 3, demonstrating a clear indication of NDO and urinary incontinence. Elevated EUS EMG activation was also observed during voiding contractions in these SCI animals [31]. Keung et al. found that minipigs were not able to void under propofol and fentanyl anesthesia before or after SCI; therefore, they utilized either an unsedated or sedation/reversal paradigm using dexmedetomidine and atipamezole to assess LUT function [31].

The pig model has also been used to explore the effects of sacral neuromodulation (SNM) on bladder function [141], demonstrating that SNM from 1–16 weeks after SCI can improve bladder function with better capacities, can lower detrusor pressures and can avoid the emergence of DSD. Moreover, the pig has been used as an animal model for the preclinical evaluation of novel cell-based therapies that aim to strengthen the urethral pressure profile as a treatment for male stress urinary incontinence following sphincter damage [200,201].

The porcine model (and in particular the Yucatan minipig) has received more recent attention in the SCI field due to pigs’ anatomical and physiological similarities to humans [22]. The extensive care required by these animals post-SCI and their complex housing needs are important considerations that definitely limit the widespread use of this model. Pigs cannot produce a micturition reflex under propofol anesthesia [31,141], thus requiring awake, mild or sedation/reversal protocols along with thorough animal training to ensure the animals are comfortable during urodynamic assessment. After SCI, most Yucatan minipigs exhibit voiding and storage dysfunction similar to clinical NLUTD, and demonstrate NDO during cystometric assessment at 12 weeks post-injury [31]. Their larger size does permit the testing of novel devices and therapies with promising results regarding sacral neuromodulation shortly after SCI [141].

## 5. Considerations for the Application of Animal Models in NLUTD Research

The species of animal selected for neurological research is often determined by identifying which aspects of the human condition the study intends to examine and the extent to which the model can reproduce the given pathology under investigation. As model systems, the different animal models are intended to simulate one or more aspects of human biology or physiology. Taking into account the advantages and disadvantages of each and how well the specific research question can be addressed with a given animal model is critical. Beyond the basic biologic and physiologic aspects of the model, practical considerations around institutional resources/infrastructure and experimental costs are important factors contributing to the choice of animal model for an experiment.

### 5.1. Basic Functional Differences

Differences in LUT anatomy and physiology exist between animals and humans and can influence decision making. For example, the rat bladder does not contain intramural ganglia [202] and ATP is a major contributor to detrusor contraction in rats. In contrast, acetylcholine is the primary activation signal in human bladders [203,204,205]. This would potentially complicate the translation of therapeutic agents acting on these systems in rats.

In addition, normal mice and rats with intact spinal cords demonstrate an increase in EUS EMG activity before the onset of voiding and phasic patterns of activity during voiding, including active and silent periods (i.e., bursting), which during voiding generate high-frequency intravesical pressure oscillations in the cytometric trace [36]. Phasic bursting EUS EMG activity has also been seen in other animals, including dogs [139,187] and some nonhuman primates [206,207]. However, EUS EMG activity is silenced during micturition in humans, pigs [31] and felines [208,209,210]; therefore, intravesical pressure oscillations with concomitant urethral sphincter activation indicates the presence of DSD after SCI. The fact that this potentially pathologic pattern of bladder and EUS activation in humans is a feature of normal voiding in mice, rats, dogs and NHPs complicates the use of these animal models for studying DSD; nonetheless, since the pattern of EUS EMG activation in DSD is significantly different from that of phasic bursting during voiding, it remains viable to use rodents for neurogenic bladder investigations with this understanding.

### 5.2. Technical and Practical Considerations

Rats and mice are the most widely available and economical animals to use, with virtually all biomedical research facilities having the infrastructure required to conduct experiments. Their small size presents some technical challenges when conducting volume and flow-related studies. For instance, while some mouse cystometric studies have estimated the flow rate during voiding [181,211], the current assessment of urine flow in mice commonly depends on the observation of urine output and increased bladder pressure, thereby hindering the ability to distinguish coordinated voiding from non-voiding urine flow (i.e., involuntary urine expulsion associated with detrusor overactivity). Although transurethral catheterization does not appear to affect urethral function, as assessed during rodent CMG [212], small PE-50 catheters are prone to blockage [212]. In addition, longitudinal assessment is more challenging with the chronic catheter and electrode implantation commonly performed in small animals (e.g., inflammation, urinary tract infection, stone formation, bleeding, and catheter blockade) [158,171] compared to transurethral approaches. To address this, a serial transurethral cystometric technique has been proposed and may offer a reliable method for longitudinal assessment [213].

Larger animal models (e.g., sheep, pig, dog, cat) are more costly for conducting NLUTD research, particularly given the context of inducing SCI with significant neurologic impairment and providing the necessary post-injury care. Additionally, research facilities may not have the necessary infrastructure to conduct large animal experiments, or may lack personnel with specialized training to manage such animals after SCI. Housing requirements for large animals are expensive and labor intensive to maintain and often preclude their use for high throughput analysis, particularly for drug screening and testing. The significant resources needed to support large animal studies limits the widespread adoption of such animal species for NLUTD research. Ethical approval for research in large animals can be more challenging to obtain and can therefore limit the opportunity for testing.

One of the key advantages of large animal models for NLUTD, such as the pig, is that they can undergo transurethral filling cystometry using the same clinical urodynamics equipment used in human studies, and they may be large enough to accommodate the study of human-sized devices. Repeated minimally invasive transurethral catheterization also allows for the longitudinal study of NLUTD in the same animal before and after SCI. In addition, the larger bladder size enables both the molecular and histological assessment of ureteral, urethral and bladder tissue from the same animal, which is not always possible in smaller models. One of the distinct disadvantages of using these models, however, is in the availability of agents and tools, such as specific antibodies for the analysis of tissue. This presently limits the depth to which mechanistic studies can go into in large animal models until such reagents become more available. Indeed, the availability of specific antibodies may incentivize research towards rodent models. Drug development studies in large animal models require higher quantities of the test drug, which may be costly for novel therapeutic agents. On the other hand, the larger size of these models can facilitate more realistic biodistribution studies, which are important for the translation of novel agents.

Future studies investigating NLUTD in animals may seek alternative methods to assess bladder pressure without an externalized catheter. Recent studies have utilized telemetric wireless technology to measure intravesical pressure in small and large animals [214]. For instance, Mickle et al. [215] investigated the use of a stretch sensor that is external to the bladder to estimate intravesical pressures in rodents. In humans, Frainey et al. tested an intravesical pressure sensor, the “UroMonitor”, which showed a strong correlation to conventional UDS catheters [216]. Other studies have developed rigid intravesical sensors to measure bladder pressure in rabbits [217] and cats [218]. Implanted telemetric pressure sensors can be used to monitor neurogenic bladders in humans and animal models without the use of catheters, anesthetic agents or restraints, and therefore may provide a more relevant method for future bladder assessment studies.

## 6. Conclusions and Discussion

In summary, cystometry allows the study of aspects of LUT function and dysfunction before and after SCI. Bladder dysfunction in post-SCI animals shares some of the features observed in human SCI subjects, however important species-specific functional differences exist, and there are additional technical considerations that influence model selection for each experiment. The selection of the most appropriate animal model requires an in-depth knowledge of each species in order to decide upon the appropriateness of the model and the outcome measures.

Small animals such as rats and mice have been extensively characterized and have been used to test a host of therapeutic agent neuromodulation strategies and understand the mechanisms that underlie the physiology of bladder function before and after SCI [145]; however, differences in LUT size, as well as in the anatomy and physiology of humans, should be considered. Larger animals provide the opportunity to test devices sized for human use and present more similarities to human LUT physiology. Ethical requirements surrounding the use of these animals in biomedical research—felines and canines in particular—have accounted for additional challenges and should be considered. Extensive post-surgical care and housing requirements can be costly for large animals, which will therefore limit the number of facilities capable of conducting such research.

Technical considerations, such as the anesthetic, catheterization technique and filling strategy used, have been shown to alter LUT function and CMG outcomes and should be taken into consideration [219]. Since the establishment of an awake, restrained cystometric assessment strategy in rodents [153,220], most rodent studies have avoided the use of an anesthetic during LUT assessment, thus providing more physiological reports of LUT function. A sedation-reversal procedure has been demonstrated in a porcine model [31], which may translate to other large animal models used in conscious urodynamics studies. Telemetric bladder pressure sensing should also be considered, as this method negates the externalization of catheter tubing and promotes more physiologically relevant recordings of LUT function during natural diuresis without the use of an anesthetic [27,86,197,221,222].

From a translational standpoint, several points may be addressed in future research. Animal models of SCI, including completely transected, contusive and compressive models, have been used for investigating NLUTD. However, it is important to consider that NLUTD induced by contusion SCI has a different time course than that of spinal transection [63], and most patients do not have a complete transection SCI [223]. Mitsui and colleagues reported differences in the extent of supraspinal projections to the lumbosacral spinal cord between contusion and transection groups [151]. Some serotonin- and dopamine beta hydroxylase-positive fibers were preserved after spinal contusion at L6, but none were present in the transection group, suggesting that the benefits of pharmacologic treatments may be different in these two lesion models [151].

Furthermore, tests of renal function in conjunction with cystometry can help monitor the progression of chronic kidney injury; however, this is not typically studied in animal models of NLUTD after SCI. Rodriguez-Romero et al. demonstrated that the glomerular filtration rate (GFR), but not tubular secretion (TS), was impaired during the spinal shock phase after contusive SCI in rats [224]. Further, Parvin et al. demonstrated the effect of SCI on pro-inflammatory cytokine expression in the kidney at the acute and sub-chronic stages [225]; however, we found no studies investigating renal function as an outcome metric in animals after SCI or able to determine a causal link between observed LUT dysfunction and upper urinary tract deterioration. Given the appreciable need to develop therapies to preserve the upper urinary tract after SCI, future studies may consider renal assessment as a valuable outcome metric.

## Figures and Tables

**Figure 1 biomedicines-11-01539-f001:**
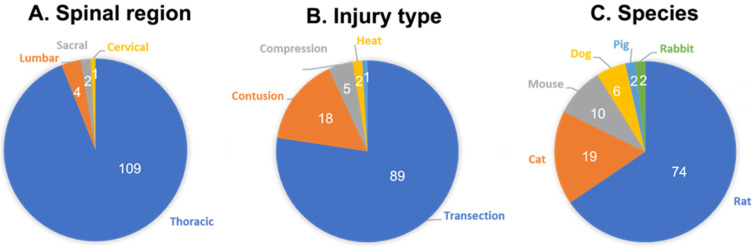
Summary of animal species, injury type and spinal level of neurogenic lower urinary tract dysfunction studies in animal models of SCI. Initial records identified through our literature search included 112 articles. Studies including two injury mechanisms, levels or species were counted twice. Values refer to the number of articles within a given category. (**A**) Among these studies, the most common spinal region studied was thoracic (109, 94%), followed by lumbar (4, 3%), sacral (2, 2%), and cervical (1, 1%). (**B**) Transection was the most common pattern of injury. Contusion and compression, characterized by compression of the spinal cord over an extended period of time, were the next most frequent injuries utilized. (**C**) Rats were the most common species among the animal models of SCI studying NLUTD, with cats and mice as second and third, respectively.

**Figure 2 biomedicines-11-01539-f002:**
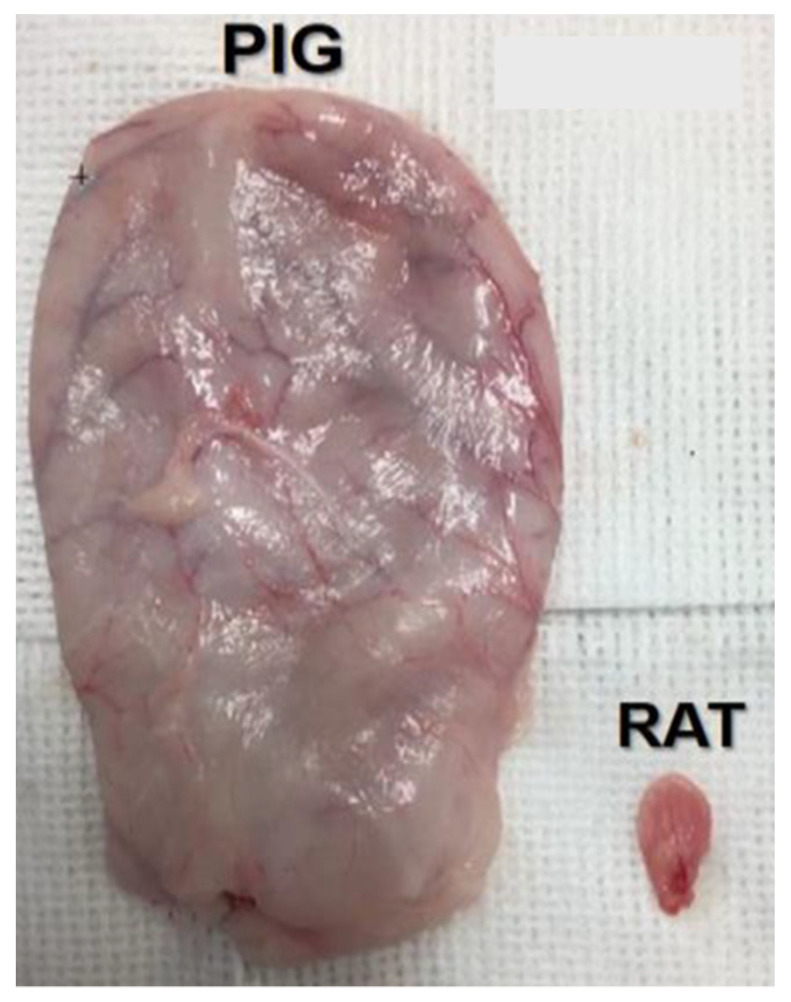
Bladder size comparison between a Yucatan minipig and a rat. A comparative assessment was made during a terminal procedure involving the bladders obtained from a Yucatan minipig and a rat.

**Figure 3 biomedicines-11-01539-f003:**
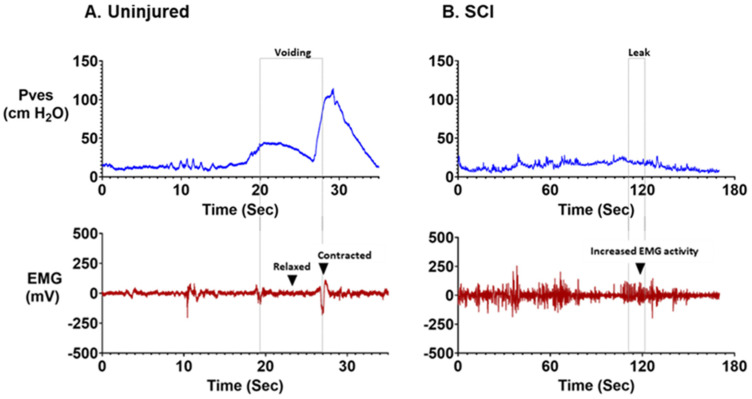
Consequences of spinal cord injury on voiding function in a large animal model of spinal cord injury (SCI) during telemetric ambulatory urodynamic monitoring. Ambulatory data recorded during natural diuresis using an implanted trans-detrusor bladder pressure monitoring system. EMG activation is recorded using bipolar electrodes implanted around the striated external urethral sphincter (EUS) muscle. The blue trace shows bladder pressure (Pves); the red trace shows EUS EMG data. (**A**) In this example of an intact control pig, one can see the bladder (vesical) pressure slowly rising, followed by voiding initiation when the pressure reaches approximately 20 cm H_2_O. (**B**) T10 contusion-compression SCI lead to the emission of urine (i.e., incontinence) after a much weaker contraction relative to pre-injury.

**Table 2 biomedicines-11-01539-t002:** Summary of animal models used to assess neurogenic lower urinary tract dysfunction before and after SCI. (ΔPdet = Maximum detrusor contraction pressure; EUS = External urethral sphincter; DSD = Detrusor sphincter dyssynergia; NDO = Neurogenic detrusor overactivity; 5′-HT = Serotonin).

Animal	What Is the Normal (i.e., Uninjured) Cystometry Pattern Established?	What Has Been Shown to Change after SCI?	What Kinds of Therapies Have Been Tested?	What Are the Disadvantages?	What Are the Advantages?	Refs.
**Rats/Mice**	EUS EMG silent and active periods (“bursting”) during voiding. Capacity: 0.3–3.8 mL Max detrusor contraction pressure (ΔPdet): 25–50 cm H_2_O Voided volume: 0.18–1.59 mL Voiding efficiency: 69–94% Contraction duration: 15–85 s	Tonic dyssynergic EUS EMG activation (DSD) Increased capacity: 0.3–26.1 Increased max detrusor ontraction pressure (ΔPdet): 25–77 cm H_2_O Increased residual volume: 0.13–4 mL Decreased voiding efficiency: 5–78% Neurogenic detrusor overactivity (NDO) present	Drugs: 5′-HT receptor agonists, adrenoreceptor agonists, inosine, n-nitrosoglutathione Cell: stem cells, Anti-Nogo-A antibodies, botulinum toxin-A Tissue: Peripheral nerve transplantation, bladder augmentation, detrusor myoplasty, hypogastric nerve resection, pelvic nerve resection. Device: Tibial, sacral and pudendal neuromodulation	Very small in comparison to human bladder. Challenges with catheterization and EMG recording EUS bursting during normal voiding complicates DSD assessment.	Low cost Widely available Easy access to reagents for imaging studies Well characterized Transgenic approaches Short lifespan	[36,40,44,53,55,56,59,67,69,73,74,75,76,77,78,79,80,81,82,84,85,86,87,88,89,90,92,93,95,97,98,101,102,103,104,105,107,108,109,110,112,115,117,118,118,119,122,150,163,164,165,168]
**Cats**	Silent EUS EMG during voiding Capacity: 5–30 mL Max bladder contraction pressure (ΔPves): 20–60 cm H_2_O Voided volume: 70 mL	Increased bladder capacity: 30–45 mL Max bladder contraction pressure (ΔPves): 20–40 cm H_2_O Decreased voided volume: 1–40 mL Reduced voiding efficiency: ~13% Detrusor areflexia immediately after SCI NDO (time-dependent; most prominent at 4 weeks post-SCI)	Stimulation: Pudendal nerve stimulation, sacral neuromodulation, deep perineal nerve stimulation, direct bladder stimulation, pelvic plexus stimulation. 5′-HT1A receptor agonists Autografting adrenal medulla	Less well characterized Challenges obtaining ethical approval More prominent health-risks during anesthesia.	Larger size	[29,30,128,129,130,131]
**Dogs**	EUS EMG bursting activity during voiding. Bladder capacity: 100–200 mL	Acute urinary retention and detrusor areflexia for 2–6 weeks post-SCI Increased capacity: 100–300 mL Decreased max bladder contraction pressure (ΔPves): 20–80 cm H_2_O NDO at 1 and 3 months post-SCI	Stimulation: Transcutaneous bladder stimulation, Pelvic nerve stimulation, Pudendal nerve stimulation, Sacral root stimulation, Cell: Matrix metalloproteinase inhibitor	Less well characterized Challenges obtaining ethical approval Increased housing cost/requirements	Larger size	[44,137,140]
**Pigs**	Silent EUS EMG during voiding. Capacity: 350–500 mL Max bladder contraction pressure (ΔPves): 15–50 cm H_2_O Voided vol: 150–800 mL Flow rate: 10–65 mL/s Efficiency: 95–100% Compliance: 40 mL/cm H_2_O	Increased capacity: 300–600 mL Decreased efficiency: 1–20% Decreased max bladder contraction pressure (ΔPves): ~27 cm H_2_O Decreased voided volume: 5–50 mL Increased residual volume: 500–700 mL Decreased compliance: 10–40 cm H_2_O Neurogenic detrusor overactivity noted in most animals at 12 weeks post-SCI Detrusor sphincter dyssynergia present.	Sacral neuromodulation Cell-based therapies including myoblast injections.	Housing costs high. Limited labs performing research. High-level care needed during recovery. Ethical challenges. Less well characterized	Clinical relevance regarding bladder size, LUT physiology, anatomy Ability to test human-sized devices Test same recording equipment as clinical cystometry	[31,141]

## Data Availability

Not applicable.

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
