# Peer review of "Assessing Neurogenic Lower Urinary Tract Dysfunction after Spinal Cord Injury: Animal Models in Preclinical Neuro-Urology Research"

_biomedicines, 2023, doi:10.3390/biomedicines11061539_

Round 1

Reviewer 1 Report

This is a very well-written review on the current animal models being used to assess neurogenic lower urinary tract dysfunction after spinal cord injury. The authors extensively describe the multitude of species that can be used, each one with their pro and cons.

The only suggestion that I would be happy to see incorporated in the manuscript is perhaps “what’s next?”. The authors nicely described some of the limitations of using animal models to study NLUD, and one of these is indeed the way to perform cystometric recordings. Especially when using rodents, the implantation of a chronic catheter is required if assessments need to be performed while the animal is awake. However, the implantation of the catheter can alter bladder morphology and function, thus alternative ways would highly warranted. As an example, although not an entirely new idea, the use of stretch sensor on the outside of the bladder wall (Mickle A.D. et al, 2019) has been investigated and looks promising to measure intravesical pressure, especially without having to artificially fill the bladder.

Here some minor comments:

·      Third paragraph of chapter 4.1 (after reference 159): “Most commonly, higher doses of agents such as [MISSING] can suppress voiding reflexes,…” - Words are missing.

·      Fifth paragraph of chapter 4.1: “… and have standardized methods for SCI induction and post-SCI care, AND, rodent models…” – the word “and,” has no context.

·      Table 2, Rats & mice: normal contraction duration of 24-83s - Isn’t this very long?

·      Table 2, Cats: “decreased max bladder contraction pressure” - This is completely the opposite of what’s observed in humans, right? In section 4.2, the authors describe the use of neuromodulatory therapies for NLUTD, mentioning that some of these might “restore bladder function after SCI”. How does the restored bladder function look like? Do these neuromodulatory therapies increase max bladder contraction pressure, so that it closely resembles the one observed in intact animals? If yes, would the authors expect an increase in max bladder contraction pressure in humans as well, even though these patients would need to reduce their max pressure?

·      Chapter 5.1: “In CONTRACT acetylcholine is the primary activation signal…” - misspelling.

Author Response

The only suggestion that I would be happy to see incorporated in the manuscript is perhaps “what’s next?”. The authors nicely described some of the limitations of using animal models to study NLUD, and one of these is indeed the way to perform cystometric recordings. Especially when using rodents, the implantation of a chronic catheter is required if assessments need to be performed while the animal is awake. However, the implantation of the catheter can alter bladder morphology and function, thus alternative ways would highly warranted. As an example, although not an entirely new idea, the use of stretch sensor on the outside of the bladder wall (Mickle A.D. et al, 2019) has been investigated and looks promising to measure intravesical pressure, especially without having to artificially fill the bladder.

First, the authors appreciate your time spent reviewing the present manuscript and your insightful comments. We appreciate the suggestion. In response, under section 5.2. Technical and practical considerations, you’ll find the addition of the following paragraph describing some of the possible next steps in the assessment of bladder function in small and large animal models including the recommended reference:

“Future studies investigating NLUTD in animals may seek alternative methods to assess bladder pressure without an externalized catheter. Recent studies have utilized telemetric wireless technology to measure intravesical pressure in small and large animals. For instance, Mickle et al. [214] investigated the use of a stretch sensor external to the bladder to estimate intravesical pressures in rodents. In humans, Frainey et al. tested an intravesical pressure sensor, the “UroMonitor”, which showed a strong correlation to conventional UDS catheters [215]. Other studies have developed rigid intravesical sensors to measure bladder pressure in rabbits [216] and cats [217]. Implanted telemetric pressure sensors can be used to monitor neurogenic bladder in humans and animal models without the use of catheters, anesthetic agents or restraint and therefore may provide a more relevant method for future bladder assessment studies.”

Table 2, Rats & mice: normal contraction duration of 24-83s - Isn’t this very long?

The data used to generate the range presented in Table 2 came from the following studies:

  • Shi et al. 2013 – Table 1. Contraction duration: 67.9±61.9s
  • Yoshiyama et al. 2000 – Table 1. Contraction duration: 37.3±3s
  • Miyazato et al. 2003 – Fig 2. Duration of isovolumetric contractions: ~1.25min
  • Cheng et al. 2004 – Table 1. Contraction duration: 0.3±0.06mins
  • Leung et al. 2007 – Table 1. Contraction duration: 16±1.2s
  • Zinck et al. 2008 – Table 2. Contraction duration: 22.3±1.9s

Given the referenced information above we conclude that the range of contraction durations in intact rats to be anywhere from 15-85s. We have revised the initial statement of 24-83s to match these findings.

Table 2, Cats: “decreased max bladder contraction pressure” - This is completely the opposite of what’s observed in humans, right? In section 4.2, the authors describe the use of neuromodulatory therapies for NLUTD, mentioning that some of these might “restore bladder function after SCI”. How does the restored bladder function look like? Do these neuromodulatory therapies increase max bladder contraction pressure, so that it closely resembles the one observed in intact animals? If yes, would the authors expect an increase in max bladder contraction pressure in humans as well, even though these patients would need to reduce their max pressure?

We acknowledge the use of neuromodulation techniques as described in section 4.2. are different in feline studies relative to those generally performed in human SCI subjects. To address this, we’ve added context to the cited studies such as Yoo et al. [178] and Gu et al. [134]. The sentences have been revised to:

“Yoo et al. found that selective stimulation of the different branches of the pudendal nerve could evoke excitatory bladder contractions after SCI [186]”

“Further, Guo et al. found that pudendal neuromodulation using a wearable stimulation device could improve bladder outcomes after SCI in cats by reducing intravesical pressure and increasing voiding efficiency 14-weeks after injury [135].”

We also revised the statement regarding the “restoration of bladder function” to improve the clarity of this message. 

“Studies such as these are informative and may facilitate translation of neuromodulation therapies to elicit or inhibit bladder contractions in SCI individuals.”

Reviewer 2 Report

This manuscript introduces the animal models of spinal cord injury comprehensively. Indeed, this is an interesting article and well written. There are some points should be revise.

1.     The rationale of setting up an SCI animal model should be reinforced and re-written. For example, we always tried to understand the pathophysiology of neurogenic bladder and tried to treat or care such patients. For this purpose, the authors should introduce current guidelines of caring SCI patients in the first paragraph. (Chen YJ, et al. Urol Sci 2023, 34; 3-9.) The first and second references are too old to be cited.

2.     A purpose to establish a reliable animal model is for studying the underlying molecular pathophysiologies. We all respect the Godfather of the bladder, prof, de Groat. However, the references published in the last century should be updated. For classic concepts and theories, the authors could reference the recent work of the Pittsburg team, such as Wada et al. Urol Sci. 2022;33(3):101-113. And, for molecular mechanisms, the author could reference another work, Shimizu et al, Int J Mol Sci, 2023, 24, 7885.

3.     This manuscript contained a maximum of older references (e.g., Ref 1, 2, 36, 41, 42, 47, 48, 49, 55, 56, 58……), so it is suggested to quote the literatures in the last three to five years, precisely. All references more than 10 years should reconsider the suitable in the modern article.  

Author Response

This manuscript introduces the animal models of spinal cord injury comprehensively. Indeed, this is an interesting article and well written. There are some points should be revise. 

  1. The rationale of setting up an SCI animal model should be reinforced and re-written. For example, we always tried to understand the pathophysiology of neurogenic bladder and tried to treat or care such patients. For this purpose, the authors should introduce current guidelines of caring SCI patients in the first paragraph. (Chen YJ, et al. Urol Sci 2023, 34; 3-9.) The first and second references are too old to be cited.

The authors appreciate your time spent reviewing the present manuscript and your insightful comments. In the revised version, the current guidelines surrounding the care of patients living with NLUTD after SCI are cited in the first paragraph of section 1 as requested:

“Accurate and reliable assessment techniques are therefore necessary to characterize NLUTD in individuals with SCI and guide their bladder management [8].”

As well as the second paragraph:

“Therefore, prompt detection of high-risk features and timely intervention, treatment and management are crucial [8].”

In addition, a sentence was added in the second paragraph of section 1 and included the reference suggested by the reviewer in the following comment (Shimizu et al. 2023), stating:

“Animal models are therefore critical for investigating in-vivo consequences of NLUTD, and they allow for the development and testing of novel therapeutic approaches and understanding the complex  pathophysiology of neurogenic bladder [20].”

Regarding the first and second references (ie. Anderson et al. 2004; Bloemen-Vrencken et al. 2005), although these studies can be seen as dated, they represent seminal surveys in the field of SCI and were among the first papers to demonstrate the need for additional focus on the effects of LUTD on SCI subjects. It is for this reason that the authors believe we should keep these references in addition to those recommended by the reviewer.  

  1. A purpose to establish a reliable animal model is for studying the underlying molecular pathophysiologies. We all respect the Godfather of the bladder, prof, de Groat. However, the references published in the last century should be updated. For classic concepts and theories, the authors could reference the recent work of the Pittsburg team, such as Wada et al. Urol Sci. 2022;33(3):101-113. And, for molecular mechanisms, the author could reference another work, Shimizu et al, Int J Mol Sci, 2023, 24, 7885.

References recommended by the reviewer have been added to the appropriate sections of the revised manuscript. For instance, Shimizu et al. 2023 is discussed in the Background section as stated above. Wada et al. 2022 is cited in the discussion paragraph stating:

“Small animals such as rats and mice have been extensively characterized and have been used to test a host of therapeutic agents neuromodulation strategies and understand the mechanisms underlying the physiology of bladder function before and after SCI [145]”

In addition, references such as Weld et al. 2000 have been replaced by Groen et al. 2016 in section 1 paragraph 2:

               “NDO with or without DSD is a common urological complication following SCI [15].”

  1. This manuscript contained a maximum of older references (e.g., Ref 1, 2, 36, 41, 42, 47, 48, 49, 55, 56, 58……), so it is suggested to quote the literatures in the last three to five years, precisely. All references more than 10 years should reconsider the suitable in the modern article.  

The authors agree that more recent publications are certainly warranted where possible; however, we believe the studies presented herein are used to summarize all of the work performed assessing NLUTD in the various animal models and are therefore important for the present review. In addition to those references recommended by the reviewer, the following studies have been updated with their more suitable modern articles.

  • 35- van Gool et al. 1979 -> Abdelkhalek et al. 2021
  • 144- de Groat et al. 1995 -> Wada et al. 2022
  • 148- Lemack et al. 1999 -> Petrosyan et al. 2023
